METHODS AND RESOURCES

# RefPlantNLR is a comprehensive collection of experimentally validated plant disease resistance proteins from the NLR family

**Jiorgos Kourelis**, **Toshiyuki Sakai**[¤a], **Hiroaki Adachi**[¤b], **Sophien Kamoun***

The Sainsbury Laboratory, University of East Anglia, Norwich Research Park, Norwich, United Kingdom

¤a Current address: Laboratory of Crop Evolution, Graduate School of Agriculture, Kyoto University, Mozume, Muko, Kyoto, Japan
¤b Current address: Graduate School of Biological Sciences, Nara Institute of Science and Technology, Ikoma, Japan
* sophien.kamoun@tsl.ac.uk

**Data Availability Statement:** Up to date versions of RefPlantNLR can be accessed via Zenodo at http://doi.org/10.5281/zenodo.3936022. This project is part of the OpenPlantNLR community on

## Abstract

Reference datasets are critical in computational biology. They help define canonical biological features and are essential for benchmarking studies. Here, we describe a comprehensive reference dataset of experimentally validated plant nucleotide-binding leucine-rich repeat (NLR) immune receptors. RefPlantNLR consists of 481 NLRs from 31 genera belonging to 11 orders of flowering plants. This reference dataset has several applications. We used RefPlantNLR to determine the canonical features of functionally validated plant NLRs and to benchmark 5 NLR annotation tools. This revealed that although NLR annotation tools tend to retrieve the majority of NLRs, they frequently produce domain architectures that are inconsistent with the RefPlantNLR annotation. Guided by this analysis, we developed a new pipeline, NLRtracker, which extracts and annotates NLRs from protein or transcript files based on the core features found in the RefPlantNLR dataset. The RefPlantNLR dataset should also prove useful for guiding comparative analyses of NLRs across the wide spectrum of plant diversity and identifying understudied taxa. We hope that the RefPlantNLR resource will contribute to moving the field beyond a uniform view of NLR structure and function.

## Introduction

Reference datasets are critical in computational biology [1,2]. They help define canonical biological features and are essential to benchmarking studies. Reference datasets are particularly important for defining the sequence and domain features of gene and protein families. Despite this, curated collections of experimentally validated sequences are still lacking for several widely studied gene and protein families. One example is the nucleotide-binding leucine-rich repeat (NLR) family of plant proteins. NLRs constitute the predominant class of disease resistance (R) genes in plants [3–5]. They function as intracellular receptors that detect pathogens and activate an immune response that generally leads to disease resistance. NLRs are thought

Zenodo: https://zenodo.org/communities/openplantnlr.

**Funding:** This work has been supported by Gatsby Charitable Foundation (https://www.gatsby.org.uk/) (TS, SK), Biotechnology and Biological Sciences Research Council (BBSRC BB/P012574 (Plant Health ISP)) (SK), European Research Council (grant number 743165, https://cordis.europa.eu/project/id/743165) (SK), Japan Society for the Promotion of Plant Science Postdoctoral fellowship (HA), and BASF Plant Science (JK, SK). The funders had no role in study design, data collection and analysis, decision to publish, or preparation of the manuscript.

**Competing interests:** I have read the journal's policy and the authors of this manuscript have the following competing interests: The authors receive funding from industry on NLR biology.

**Abbreviations:** ANK, ankyrin; ARM, armadillo; CC, coiled-coil; CDS, coding sequence; C-JID, C-terminal jelly roll/Ig-like domain; ETI, effector-triggered immunity; LRR, leucine-rich repeat; NACHT, neuronal apoptosis inhibitory protein, MHC class II transcription activator, HET-E incompatibility locus protein from *Podospora anserina*, and telomerase-associated protein 1; NB-ARC, nucleotide-binding adaptor shared by APAF-1, certain R gene products, and CED-4; NLR, nucleotide-binding leucine-rich repeat; NOD, nucleotide-binding and oligomerization domain; SSFR, superstructure-forming repeat; TIR, Toll/interleukin-1 receptor; TPR, tetratricopeptide repeat.

to be engaged in a coevolutionary tug-of-war with pathogens and pests. As such, they tend to be among the most polymorphic genes in plant genomes, both in terms of sequence diversity and copy number variation [6]. Ever since their first discovery in the 1990s, hundreds of NLRs have been characterized and implicated in pathogen and self-induced immune responses [4]. NLRs are among the most widely studied and economically valuable plant proteins, given their importance in breeding crops with disease resistance [7].

NLRs occur widely across all kingdoms of life where they generally function in non-self-perception and innate immunity [3,8,9]. In the broadest biochemical definition, NLRs share a similar multidomain architecture consisting of a nucleotide-binding and oligomerization domain (NOD) and a superstructure-forming repeat (SSFR) domain [10]. The NOD is either an NB-ARC (nucleotide-binding adaptor shared by APAF-1, certain R gene products, and CED-4) or NACHT (neuronal apoptosis inhibitory protein, MHC class II transcription activator, HET-E incompatibility locus protein from *Podospora anserina*, and telomerase-associated protein 1), whereas the SSFR domain can be formed by ankyrin (ANK) repeats, tetratricopeptide repeats (TPRs), armadillo (ARM) repeats, WD repeats, or leucine-rich repeats (LRRs) [10,11]. Plant NLRs exclusively carry an NB-ARC domain with the C-terminal SSFR consisting typically of LRRs (Fig 1A). The NB-ARC domain has been used to determine the evolutionary relationships between plant NLRs, given that it is the only domain that produces reasonably good global alignments across all members of the family. In flowering plants (angiosperms), NLRs form 3 main monophyletic groups with distinct N-terminal domain fusions: the TIR-NLR subclade containing an N-terminal Toll/interleukin-1 receptor (TIR) domain, the CC-NLR-subclade containing an N-terminal Rx-type coiled-coil (CC) domain, and the $CC_R$-NLR subclade containing an N-terminal RPW8-type CC ($CC_R$) domain [12]. Additionally, Lee and colleagues [13] have recently proposed that the G10-subclade of NLRs is a monophyletic group containing a distinct type of CC (here referred to as $CC_{G10}$; $CC_{G10}$-NLR). NLRs also occur in nonflowering plants where they carry additional types of N-terminal domains such as kinases and α/β hydrolases [11].

Plant NLRs likely evolved from multifunctional receptors to specialized receptor pairs and networks [14,15]. NLRs that combine pathogen detection and immune signaling activities into a single protein are referred to as "functional singletons," whereas NLRs that have specialized in pathogen recognition or immune signaling are referred to as "sensor" or "helper" NLRs, respectively. About one-quarter of NLR genes occur as "genetic singletons" in plant genomes, whereas the others form genetic clusters often near telomeres [16]. This genomic clustering likely aids the evolutionary diversification of this gene family and subsequent emergence of pairs and networks [6,15]. The emerging picture is that NLRs form genetic and functional receptor networks of varying complexity [15,17].

The mechanism of pathogen detection by NLRs can be either direct or indirect [4]. Direct recognition involves the NLR protein binding a pathogen-derived molecule or serving as a substrate for the enzymatic activity of a pathogen virulence protein (known as effectors). Indirect detection is conceptualized by the guard and decoy models where the status of a host component—the guardee or decoy—is monitored by the NLR [18,19]. Some sensor NLRs known as NLR-IDs contain noncanonical "integrated domains" that can function as decoys to bait pathogen effectors and enable pathogen detection [20–22]. These extraneous domains appear to have evolved by fusion of an effector target domain into an NLR [20,21,23]. The sequence diversity of integrated domains in NLR-IDs is staggering, indicating that novel domain acquisitions have repeatedly occurred throughout the evolution of plant NLRs [21,24].

Given their multidomain nature, sequence diversity, and complex evolutionary history, prediction of NLR genes from plant genomes is challenging. Several bioinformatic tools have been developed to extract plant NLRs from sequence datasets. As an input, these tools take

**A)**

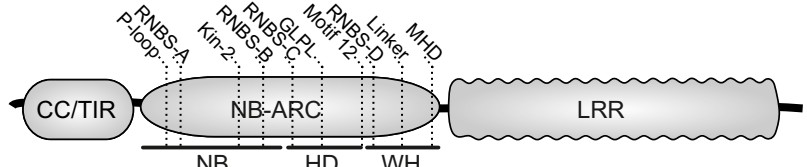

**Fig 1. Number of experimentally validated RefPlantNLR sequences per plant genus.** (**A**) Domain architecture of typical plant NLRs. The structural features and conserved motifs of the NB-ARC are indicated. (**B**) The number of experimentally validated NLRs per plant genus ($N$ = 481), and (**C**) the per genus reduced redundancy set at a 90% sequence similarity threshold ($N$ = 303) are plotted as a stacked bar graph. (**D**) The class of pathogen to which NLRs in the RefPlantNLR dataset confer a response. Some NLRs may be involved in the response against multiple classes of pathogens, while others have a helper role or are found to be involved in allelic variation in autoimmune/hybrid necrosis responses, and (**E**) the per genus reduced redundancy set at a 90% sequence similarity threshold are plotted as a stacked bar graph. The number of experimentally validated NLRs belonging to the monophyletic TIR-NLR, CC-NLR, $CC_R$-NLR, or $CC_{G10}$-NLR subclade members is indicated. Underlying data and R code to reproduce the figures in **S5 Data**. CC, coiled-coil; HD, helical domain of apoptotic protease-activating factors; LRR, leucine-rich repeat; NB, P-loop containing NTPase domain; NLR, nucleotide-binding leucine-rich repeat; TIR, Toll/interleukin-1 receptor; WD, winged helix domain.

either annotated genomic features and transcriptomic data or alternatively can be run directly on the unannotated genomic sequence. NLR-Parser, RGAugury, RRGPredictor, and DRAGO2 identify transcript and protein sequences that have features of NLRs and are best described as NLR extractors [25–28]. RGAugury, RRGPredictor, and DRAGO2 also extract other classes of immune-related genes in addition to NLRs. These various tools use predefined motifs to classify sequences as NLRs, but they differ in the methods and pipelines. NLR-Annotator—an extension of NLR-Parser—and NLGenomeSweeper can also use unannotated genome sequences as input to predict the genomic locations of NLRs [29,30]. This output then requires manual annotation to extract the final gene models, and some of the annotated loci may represent partial or pseudogenized genes.

The goal of this study is to provide a curated reference dataset of experimentally validated plant NLRs. This version of RefPlantNLR (v.20210712_481) consists of 481 NLRs from 31 genera belonging to 11 orders of flowering plants. We used RefPlantNLR to determine the canonical features of functionally validated plant NLRs and benchmark NLR extraction tools. We found that these NLR extraction tools can extract the majority of NLRs in the RefPlantNLR dataset; however, the domain architecture analysis produced by these tools is often inconsistent with that of RefPlantNLR. In order to simplify NLR extraction, functional annotation, and phylogenetic analysis, we developed NLRtracker: a pipeline that uses InterProScan [31] and predefined NLR motifs [32] to extract NLRs and provide domain architecture analyses based on the canonical features found in the RefPlantNLR dataset. Additionally, NLRtracker outputs the extracted NB-ARC domain facilitating downstream phylogenetic analysis. RefPlantNLR should also prove useful in guiding comparative and phylogenetic analyses of plant NLRs and identifying understudied taxa for future studies.

## Results and discussion

### Construction of the RefPlantNLR dataset

To construct the current version of RefPlantNLR (v.202110712_481, **S1–S3 Dataset**), we manually crawled through the literature, extracting plant NLRs that have been experimentally validated to at least some degree. We defined experimental validation broadly as genes reported to be involved in any of the following: (1) disease resistance; (2) disease susceptibility, including effector-triggered immune pathology or trailing necrosis to viruses; (3) hybrid necrosis; (4) autoimmunity; (5) NLR helper function or involvement in downstream immune responses; (6) negative regulation of immunity; and (7) well-described allelic series of NLRs with different pathogen recognition spectra even if not reported to confer disease resistance. We defined NLRs as sequences containing the NB-ARC domain (Pfam signature PF00931) or a P-loop containing nucleoside triphosphate hydrolases (NTPase) domain (SUPERFAMILY signature SSF52540) combined with plant-specific NLR motifs [32] (see Material and methods for the used motifs) (**Fig 1A**). This resulted in 479 sequences. We also included RXL [33], which has an N-terminal Rx-type CC domain and C-terminal LRR domain, as well as *At*NRG1.3 [34], which has a C-terminal LRR domain, both of which contain the RNBS-D motif of the NB-ARC domain but otherwise do not get annotated with a P-loop containing NTPase domain. Altogether, these 481 sequences form the current version of RefPlantNLR (**S1 Table**).

In addition to the 481 NLRs present in this version of RefPlantNLR, we separately collected several characterized animal, bacterial, and archaeal NB-ARC proteins (**S2 Table**, **S4 Dataset**), which can be used as outgroups for comparative analyses. Furthermore, several characterized plant immune components have features often found in NLRs—such as the RPW8-type CC or the TIR domain—but lack the NB-ARC domain or NB-ARC–associated motifs that we used to define NLRs (see above). Since these proteins may have common origins with plant NLRs or

may be useful for comparative analysis of these domains, we have collected them separately as well (S3 Table, S5–S7 Dataset).

## Description of the RefPlantNLR dataset

The 481 RefPlantNLR entries belong to 31 genera of flowering plants (Fig 1B and 1C) and are described in S1 Table. The description includes amino acid, coding sequence (CDS) and locus identifiers, as well as the organism from which the NLR was cloned, the article describing the identification of the NLR, the pathogen type and pathogen to which the NLR provides resistance (when applicable), the matching pathogen effector, additional host components required for pathogen recognition (guardees or decoys) or required for NLR function, and the articles describing the identification of the pathogen and host components. From this dataset, we extracted 472 unique NLRs and 488 NB-ARC domains of which 406 were unique (S8 and S9 Dataset). NLRs with identical amino acid sequences were recovered because they have different resistance spectra when genetically linked to different sensor NLR allele (e.g., alleles of Pik), are different in noncoding regions leading to altered regulation (e.g., RPP7 alleles), or have been independently discovered in different plant genotypes (e.g., RRS1-R and SLH1).

The distribution of the RefPlantNLR entries across plant species mirrors the most heavily studied taxa, i.e., *Arabidopsis*, Solanaceae (*Solanum*, *Capsicum*, and *Nicotiana*), and cereals (*Oryza*, *Triticum*, and *Hordeum*) (Fig 1B). These 7 genera comprise 77% (370 out of 481) of the RefPlantNLR sequences. When accounting for redundancy by collapsing similar sequences (>90% overall amino acid identity per genus), these 7 genera would still account for 73% (220 out of 303) sequences (Fig 1C). It should be noted that there could be different evolutionary rates between NLRs, and, hence, some subfamilies may still be overrepresented in the reduced redundancy set.

In total, 31 plant genera representing 11 taxonomic orders are listed in RefPlantNLR. Interestingly, these species represent a small fraction of plant diversity with only 11 of 59 major seed plant (spermatophyte) orders described by Smith and Brown represented, and not a single entry from nonflowering plants (S4 Table) [35]. *Arabidopsis* remains the only species with experimentally validated NLRs from the 4 major clades (CC-NLR, $CC_{G10}$-NLR, $CC_R$-NLR, and TIR-NLR) (Fig 1). For *Arabidopsis*, tomato, and rice, we compared the distribution of NLRs across the 4 major clades in the RefPlantNLR dataset and the published genome and found no major differences (S1A Fig).

We also mapped the frequency of the pathogens that are targeted by RefPlantNLR entries. Most validated NLRs in the RefPlantNLR dataset are involved in responses against fungi followed by oomycetes (Fig 1D and 1E for the reduced redundancy set). Responses to certain pathogen taxa is not constrained to particular subclasses of NLRs as all of TIR-NLRs, $CC_{G10}$-NLRs, and CC-NLRs are involved in resistance to the main pathogen classes (fungi, oomycete, bacteria, and viruses). The notable exception is the $CC_R$-NLR subclade, which has only been validated for its helper function (Fig 1D and 1E). Additionally, $CC_{G10}$-NLR subclade members have not been assigned a helper activity, and $CC_R$-NLR subclade members have not been implicated in autoimmunity or hybrid necrosis (Fig 1D and 1E), even though several RPW8-only proteins are involved in hybrid necrosis [36,37].

The average length of RefPlantNLR sequences varies depending on the subclass (Fig 2A and 2C for the reduced redundancy set). CC-NLRs varied from 665 to 1,845 amino acids (mean = 1,079, $N = 347$), whereas TIR-NLR varied from 380 to 2,048 amino acids (mean = 1,159, $N = 105$). NB-ARC domains were more constrained (mean = 345, $N = 406$, stdev = 33) (Fig 2B). Nonetheless, 23 atypically short NB-ARCs (155 to 274 amino acids) and 1 long NB-ARC (422 amino acids) were observed at more than 2 standard deviations of the

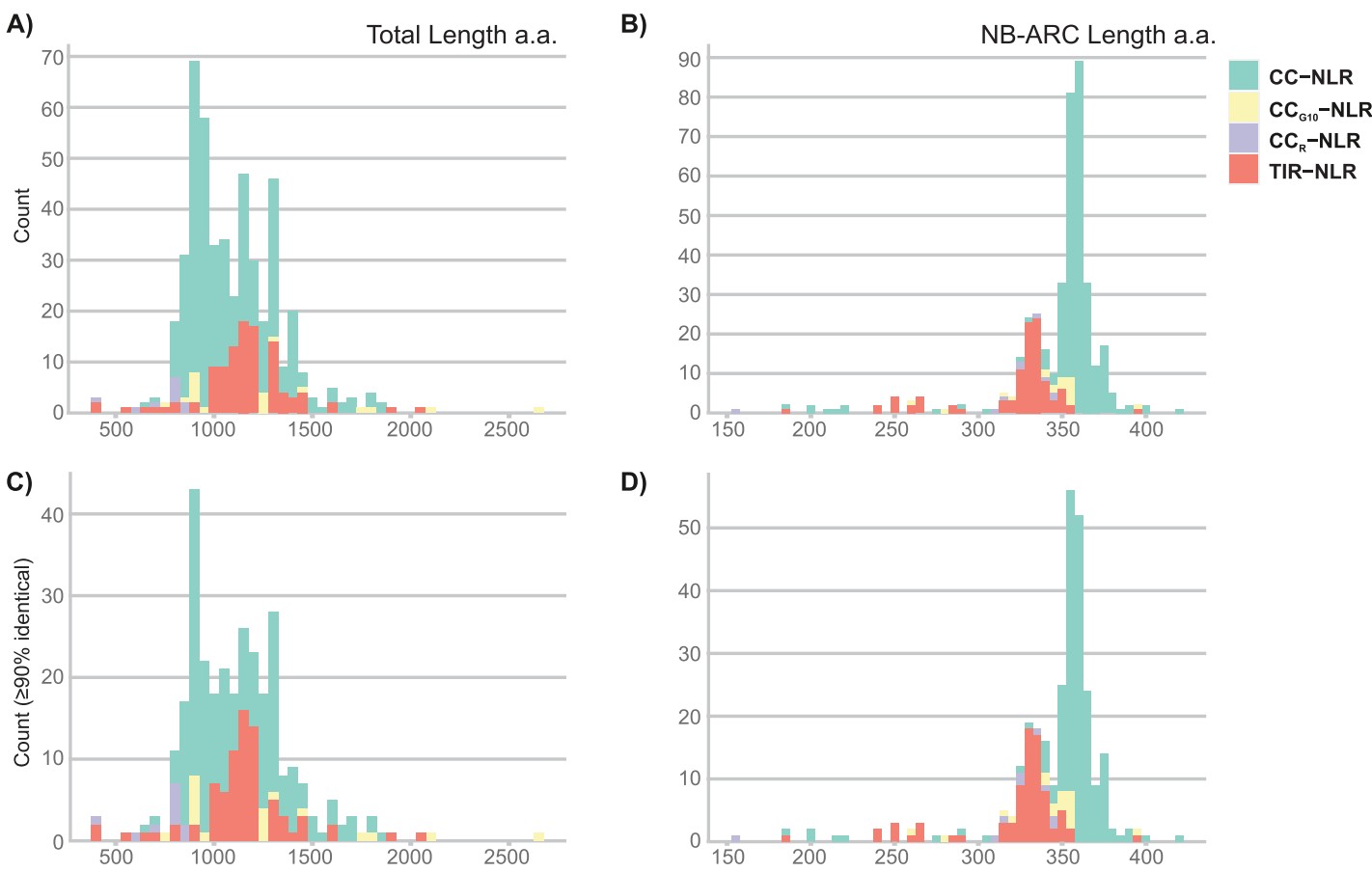

**Fig 2. Length distribution RefPlantNLR amino acid sequence and extracted NB-ARC domains.** Length distribution of the RefPlantNLR sequences. (**A**) Histogram of RefPlantNLR amino acid sequence length (binwidth 50aa, $N$ = 481). (**B**) Histogram of the unique RefPlantNLR extracted NB-ARC domain (SUPERFAMILY signature SSF52540) amino acid sequence length (binwidth 5aa, $N$ = 406). (**C**) Histogram of amino acid sequence length of the reduced redundancy RefPlantNLR set at a 90% amino acid similarity threshold (binwidth 50aa, $N$ = 303). (**D**) Histogram of the extracted NB-ARC domain from the reduced redundancy RefPlantNLR set (binwidth 5aa, $N$ = 296). Color coding according to NLR subfamily. Underlying data and R code to reproduce the figures in **S5 Appendix**. CC, coiled-coil; NB-ARC, nucleotide-binding adaptor shared by APAF-1, certain R gene products, and CED-4; NLR, nucleotide-binding leucine-rich repeat; TIR, Toll/interleukin-1 receptor.

mean illustrating the overall flexibility of plant NLRs even for this canonical domain (**Fig 2B and 2D** for the reduced redundancy set).

We noted that some of the unusually small NLRs lacked an SSFR domain, while some of the small NB-ARC domains appeared to be partial duplications of this domain. In order to look at domain architecture of NLRs more widely and to determine whether these unusual features are common, we functionally annotated the RefPlantNLR dataset using InterProScan [31] and predefined NLR motifs [32], as well as using LRRpredictor [38] (**S10 Dataset**) and an HMM for the recently discovered C-terminal jelly roll/Ig-like domain (C-JID) of TIR-NLRs [39] (**S11** and **S12 Dataset** for the combined GFF annotation). This functional annotation can be visualized using the refplantnlR R package. We used this functional annotation to map the domain architecture of RefPlantNLR proteins (**Fig 3A and 3B** for the reduced redundancy set).

Even though CC-NLR and TIR-NLR domain combinations were the most frequent (61% and 19%, respectively), we observed additional domain combinations. In the RefPlantNLR dataset, a subset of NLRs lack the N-terminal domain but still group with the major NLR clades based on the NB-ARC phylogeny. Some TIR-NLRs lack an SSFR domain. Noncanonical

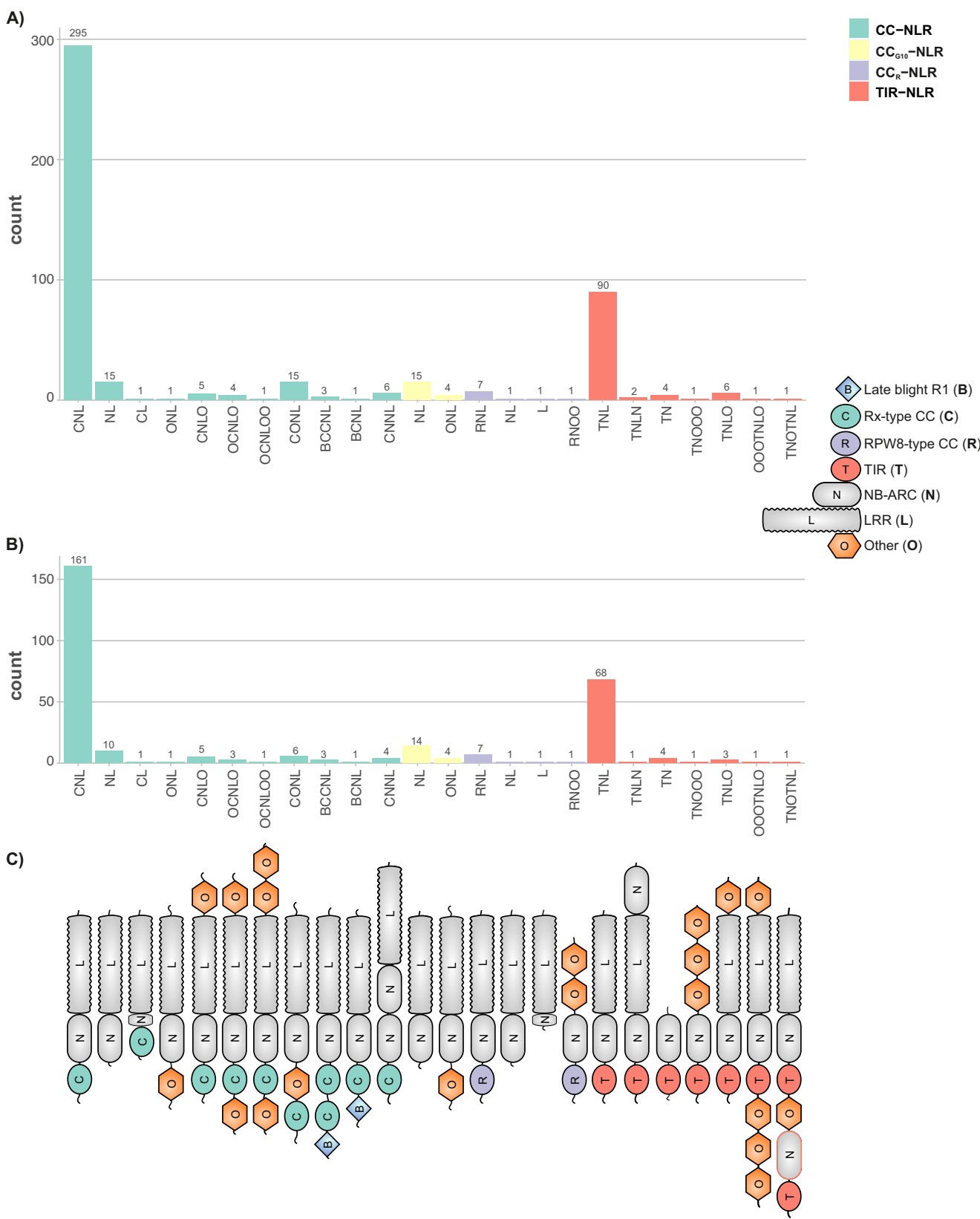

**Fig 3. Domain architecture of the RefPlantNLRs.** Bar chart of the domain architecture of (**A**) RefPlantNLRs (*N* = 481), or (**B**) the per genus reduced redundancy RefPlantNLR set at an overall 90% amino acid similarity per genus (*N* = 303). **C)** Schematic representation of domain architecture. Used InterPro signatures for each of the domains are highlighted in the Material and methods. There is currently no InterProScan signature or motif for the CC$_{G10}$ N-terminal domain. Underlying data and R code to reproduce the figures in **S5 Appendix**. CC, coiled-coil; LRR, leucine-rich repeat; NB-ARC, nucleotide-binding adaptor shared by APAF-1, certain R gene products, and CED-4; NLR, nucleotide-binding leucine-rich repeat; TIR, Toll/interleukin-1 receptor.

integrated domains are found in all NLR subfamilies and occur at the N-terminus, in between the N-terminal domain and the NB-ARC domain, at the C-terminus, or both ends. Of these noncanonical domains, the N-terminal late-blight resistance protein R1 domain (also known as the Solanaceae domain; Pfam signature PF12061) only occurs in association with the NB-ARC domain and has an ancient origin likely in the most recent common ancestor of the Asterids and Amaranthaceae [40]. Other noncanonical domains are also more widespread, including the monocot-specific integration of a zinc-finger BED domain in between the CC and NB-ARC domain [41,42]. Finally, some NLRs have significantly truncated NB-ARC domains as is the case for Pb1, *At*NRG1.3, and RXL (**Fig 3C**). For *Arabidopsis* and rice, the number of characterized NLRs containing integrated domains appears to be slightly enriched as compared to all NLRs in the reference genome, whereas there is no NLRs with integrated domains identified in the tomato reference genome (**S1B Fig**). Finally, in *Arabidopsis*, there remains a number of NLR domain architectures, which have no counterpart in the RefPlantNLR set (**S1C Fig**).

We explored the phylogenetic diversity of RefPlantNLR proteins using the extracted NB-ARC domains with non-plant NB-ARC domains as an outgroup (**Fig 4**, **S13–S15 Dataset**). As with previously reported NLR phylogenetic analyses, RefPlantNLR sequences generally grouped in well-defined clades, notably CC-NLR, CC$_{G10}$-NLR, CC$_R$-NLR, and TIR-NLR. Within this phylogeny, some of the branches, notably of Wed and Pi54, are long and may represent highly diverged NB-ARC domains. Since Pb1 [43], RXL [33], and *At*NRG1.3 [34] do not match the Pfam NB-ARC domain, they were not included in this phylogenetic analysis.

## Benchmarking NLR annotation tools using RefPlantNLR

We took advantage of the RefPlantNLR dataset to benchmark NLR annotation tools by determining their sensitivity in retrieving NLRs and accuracy in annotating NLR domain architecture. This is particularly justified because the majority of NLR prediction tools have only been evaluated using the reference *Arabidopsis* NLRome, which is not representative of NLR diversity across flowering plants (**Fig 1**). We selected 5 NLR annotation tools for benchmarking (**Table 1**). These tools differ in the methods used for NLR extraction and functional annotation. NLGenomeSweeper, RGAugury, and RRGPredictor all use InterProScan [31] to functionally annotate sequences and extract NLRs based on co-occurrences of certain domains; however, they differ in which signatures are considered for the functional annotation. By contract, DRAGO2 relies on custom HMM models to functionally annotate sequences, whereas NLR-Annotator uses MEME with custom NLR motifs [32] for NLR extraction.

Since NLR-Annotator and NLGenomeSweeper only take nucleotide sequence input, whereas RGAugury only works on protein sequences, we decided to proceed with the benchmarking using only the RefPlantNLR entries with CDS information (457/481). In this way, we ensured that we could compare the tools on the same number of sequences. Out of the NLR-extraction tools, DRAGO2 has the highest sensitivity, retrieving all of the RefPlantNLR entries when run on amino acid sequences (**Fig 5A**, **Table 1**). NLR-Annotator has the second highest sensitivity, retrieving 448/457 (98.0%) of the sequences (**Fig 5A**, **Table 1**). It has previously been noted that NLR-Annotator does not perform well on retrieving the CC$_R$-NLR subclade members [25]. Indeed, NLR-Annotator missed 7/10 (70%) of CC$_R$-NLRs in the RefPlantNLR

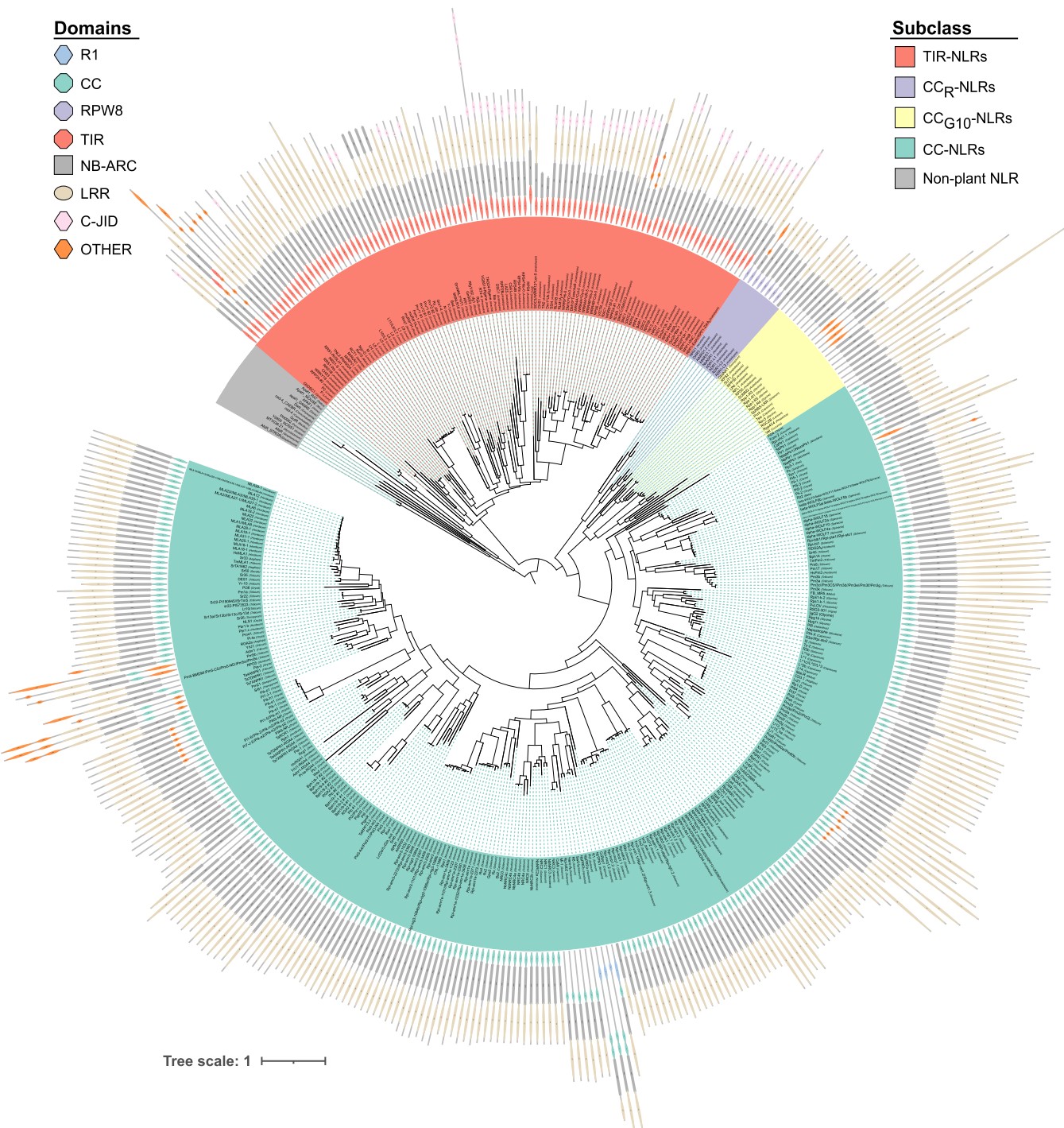

**Fig 4. Phylogenetic diversity of RefPlantNLR sequences.** The tree, based on the NB-ARC domain, was inferred using the Maximum Likelihood method based on the JTT model [44]. The tree with the highest log likelihood is shown. NLRs with identical NB-ARC domains are collapsed, while for those with multiple NB-ARC domains, the NB-ARC are numbered according to order in the protein. The tree was rooted on the non-plant NLR outgroup. The TIR-NLR, CC-NLR, CC$_R$-NLR, and CC$_{G10}$-NLR subclades are indicated. Domain architecture is shown as in Fig 3. CC, coiled-coil; C-JID, C-terminal jelly roll/Ig-like domain; JTT, Jones–Taylor–Thornton; LRR, leucine-rich repeat; NB-ARC, nucleotide-binding adaptor shared by APAF-1, certain R gene products, and CED-4; NLR, nucleotide-binding leucine-rich repeat; TIR, Toll/interleukin-1 receptor.

**Table 1. NLR annotation tools.**

| | | Output | | RefPlantNLR ($N = 429$) | |
|---|---|---|---|---|---|
| *Tool* | *Input* | *Functional annotation* | *NB-ARC* | *Sensitivity* | *Annotation specificity* |
| *DRAGO2* [27] | AA/transcripts | Coils, custom HMM models, TMHMM | No | 100%/ 99.3%* | 45.2% |
| *NLGenomeSweeper* [30] | Transcripts/Genomic | Coils, InterProScan | Yes | 98.0% 88.9%** | 31.5% 23.1%** |
| *NLR-Annotator* [29] | Transcripts/Genomic | NLR motif MEME | Yes | 98.0% 97.3%** | 88.2% 87.0%** |
| *RGAugury* [26] | AA | Coils, InterProScan, Pfam, Phobius | No | 96.9% | 61.1% |
| *RRGPredictor* [28] | AA/transcripts | Coils, InterProScan | No | 95.4% | 61.9% |
| *NLRtracker* | AA/transcripts | InterProScan, NLR motif MEME | Yes | 100% | 100% |

*AA/CDS input.

**CDS/Genomic input. Gene models were available for 407 NLRs.

CDS, coding sequence; HMM, Hidden Markov model; NB-ARC, nucleotide-binding adaptor shared by APAF-1, certain R gene products, and CED-4; NLR, nucleotide-binding leucine-rich repeat.

dataset, while it retrieved all TIR-NLRs and $CC_{G10}$-NLRs and only missed 2/326 (0.6%) of CC-NLRs (**Fig 5B**). Additionally, NLR-Annotator performs similarly on extracted genomic sequence, retrieving 396/407 (97.3%) of RefPlantNLR entries with associated genomic information (**S2 Fig**). NLGenomeSweeper, which like NLR-Annotator also takes either CDS or genomic sequence as an input, performs considerably worse on genomic input as compared to CDS input retrieving 362/407 (88.9%) of RefPlantNLR entries using extracted genomic sequence as an input versus 448/457 (98.0%) of RefPlantNLR entries when CDS was used as an input (**S2 Fig**). Both NLR-Annotator and NLGenomeSweeper duplicate NLRs with multiple NB-ARC domains, potentially artificially inflating the number of NLRs extracted.

Next, we compared sensitivity and domain annotation accuracy of the NLR annotation tools according to the 4 main NLR subclades. Since these tools only functionally annotate the canonical NLR domains, we did not consider integrated domains and the late blight R1 domain. While DRAGO2 is the most sensitive in retrieving NLRs, it correctly annotated the domain architecture of less than half (44.9%) of the RefPlantNLR sequences (**Fig 5B**, **Table 1**). DRAGO2, RGAugury, and RRGPredictor often failed to functionally annotate the CC domain (**Fig 5B**). Since these tools use Coils [45] to predict CC domains, we conclude that this program is not very sensitive to predict plant NLR CC domains. Additionally, Coils does not distinguish between the different types of CC domains such as the RPW8-type CC or Rx-type CC. Although NLR-Annotator does not automatically output a domain architecture analysis as the other tools, upon conversion of the motif analysis to domain architecture, we found that NLR-Annotator has the highest domain annotation accuracy of all tools, correctly annotating 403/457 (88.2%) of the NLRs (**Fig 5B**, **Table 1**). The other tools did not perform much better than DRAGO2, correctly annotating between 31.5% to 61.9% of RefPlantNLR entries (**Fig 5B**, **Table 1**). When looking at the different NLR subclades, it becomes clear that most tools correctly identify and annotate TIR-NLRs, while domain prediction accuracy is lower for the other NLR subclades (**Fig 5B**). The exception to this is NLR-Annotator, which accurately annotates the domains of 288/326 (88.3%) CC-NLRs. This is possibly because NLR-Annotator was validated with the wheat genome, which contains a large proportion of CC-NLRs and some of the used motifs are specific to monocot CC-NLRs [32], whereas the other tools were validated with *Arabidopsis*, which has a higher abundance of TIR-NLRs as compared to other

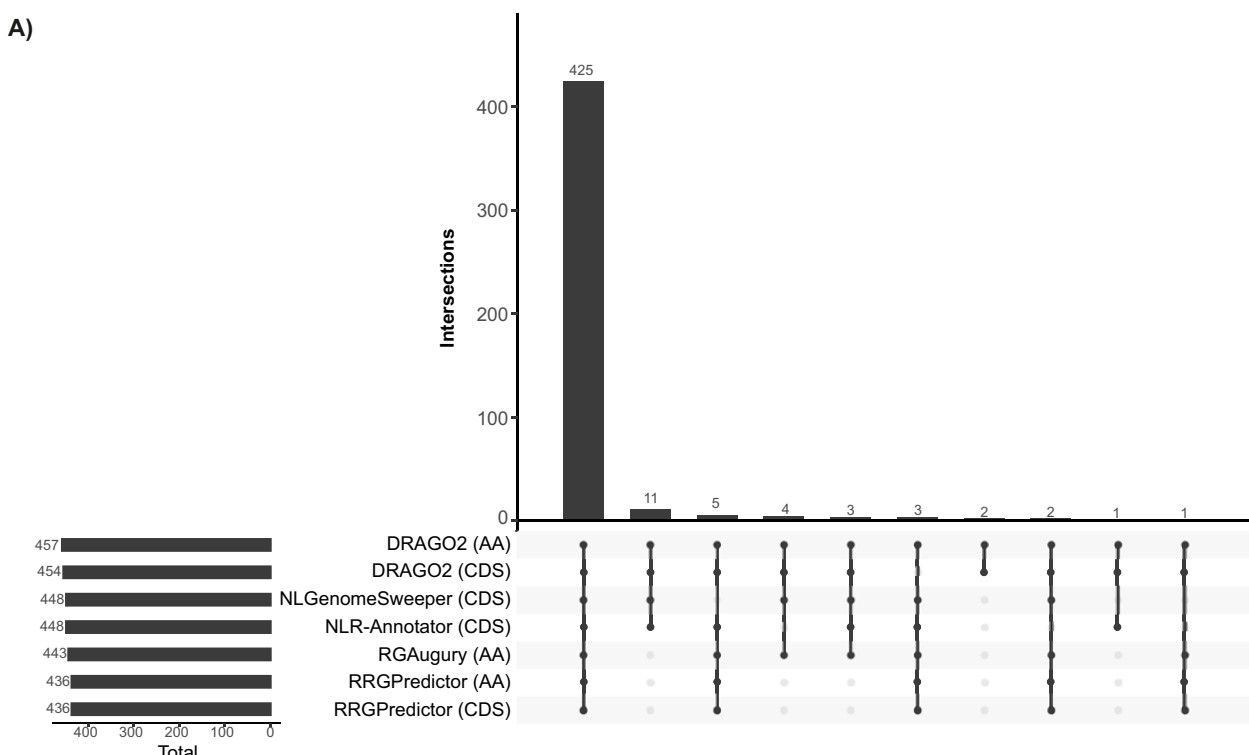

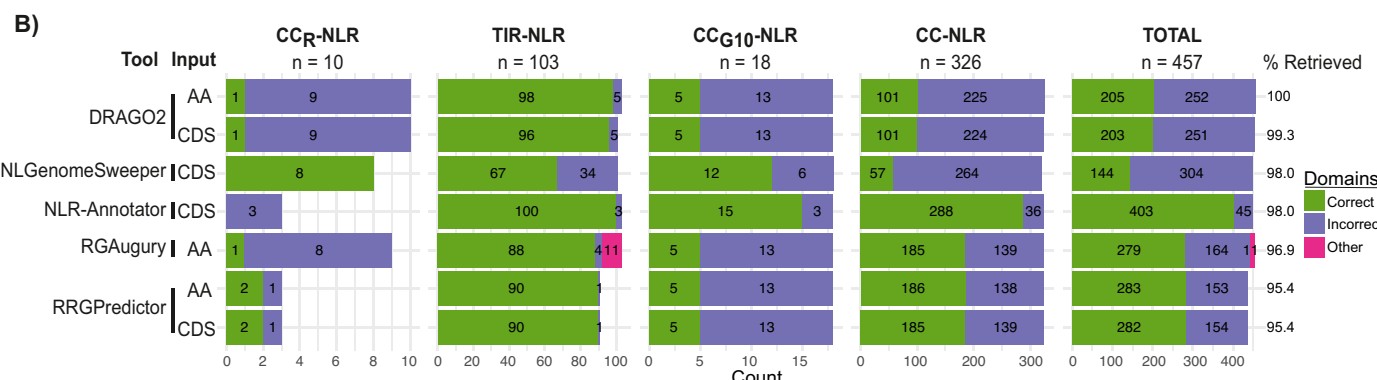

**Fig 5. Benchmarking NLR annotation tools using RefPlantNLR.** Benchmarking of NLR annotation tools using the RefPlantNLR dataset for which a CDS entry was available (*N* = 457). (**A**) UpSet plot showing intersection of RefPlantNLR entries retrieved by each annotation tool. (**B**) Domain architecture analysis produced by each NLR annotation tool per NLR subclass. Correct domain architecture is consistent with RefPlantNLR annotation, incorrect is inconsistent with RefPlantNLR annotation. Other is retrieved by NLR annotation tool but not reliably classified as NLR. Underlying data and R code to reproduce the figures in **S1 Appendix**. CDS, coding sequence; NLR, nucleotide-binding leucine-rich repeat.

species. Finally, comparing these NLR annotation tools on the reduced redundancy RefPlantNLR set revealed a similar pattern (**S3 Fig**, **S1 Appendix** for the full analysis). Based on the benchmarking using RefPlantNLR, we find DRAGO2 to be the most sensitive tool for NLR extraction, while NLR-Annotator is the most sensitive tool for use on genomic input. None of the tools performs well on the domain architecture analysis except for NLR-Annotator; however, to extract such a domain architecture output from NLR-Annotator does require a substantial effort on the user side.

## NLRtracker: An NLR extraction and annotation pipeline based on the core features of RefPlantNLR

To address the limitations of the current NLR annotation tools highlighted above, we generated a novel pipeline we called NLRtracker. NLRtracker uses InterProScan [31] and the predefined NLR motifs [32] to annotate all sequences in a given proteome or transcriptome and then extracts and annotates NLRs based on the core NLR sequence features (late blight R1, TIR, RPW8, CC, NB-ARC, LRR, and integrated domains) found in the RefPlantNLR dataset (**Fig 6A**, **S2 Appendix**, **Table 1**). The functional annotation can then be visualized using the refplantnlR R package or other software of choice. Additionally, NLRtracker extracts the NB-ARC domain for comparative phylogenetic analysis. Since NLRtracker is based on the features found in the RefPlantNLR dataset, it exactly reproduces the RefPlantNLR domain architecture and extracts all RefPlantNLR entries. To compare NLRtracker to other NLR annotation tools, we used the *Arabidopsis*, tomato, and rice RefSeq genomes. In this way, we could compare whether NLRtracker also performs well on datasets other than RefPlantNLR. In addition, we could also assess the accuracy of each NLR annotation tool, which is not possible with the RefPlantNLR dataset.

Using all tools, we extracted a total of 1,615 NLRs from the reference *Arabidopsis* ($N = 441$), tomato ($N = 250$), and rice ($N = 924$) genomes (**Fig 6B**). The total number of NLRs belonging to each subclade in each species is reflected in the RefPlantNLR dataset (**Fig 6B**). In addition to the 4 main subclades of NLRs, we also retrieved a highly conserved TIR-NB-ARC (TN) class of proteins, which phylogenetically clusters separately from all other plant NLRs and whose gene structure is clearly distinct from other TIR-NLRs [46], as well as certain NB-ARC containing proteins, which did not clearly belong to any of these clades (**S4 Fig**). This included an *Arabidopsis* NB-ARC protein with an integrated ZBTB8B domain in between the NB-ARC and the LRR, as well as a rice protein containing an NB-ARC domain with a C-terminal ARM-type SSFR (**Fig 6B**, **S3 Appendix** for the full analysis).

Using this dataset, we evaluated the sensitivity and specificity of each NLR annotation tool. Sensitivity was defined as the total percentage NLRs retrieved out of the total NLR dataset, while specificity was defined as the total number of sequences annotated as NLRs being genuine NLRs. False positives could include TIR- or RPW8-only proteins annotated as genuine NLRs, or unrelated sequences annotated as NLRs. On this dataset, NLRtracker had the highest sensitivity (retrieving 1,611/1,615 (99.8%) NLRs with 100% accuracy (**Table 2**, **Fig 6B**, **S5 Fig**). The 4 missed sequences were retrieved by DRAGO2 and included 1 NLR with an N-terminal CC-domain and a C-terminal LRR but which did not get annotated with an NB-ARC domain using InterProScan or the predefined NLR motifs, and 3 truncated proteins, 2 of which contain an LRR domain while one does not get annotated with any domain using InterProScan.

Of the preexisting tools DRAGO2 was the most sensitive, retrieving 1,526/1,615 (94.5%) NLRs; however, it also was the least accurate method extracting 91 false positives (**Table 1**, **S5 Fig**). These false positives were predominantly proteins containing a P-loop containing NTPase domain unrelated to the NB-ARC domain, e.g., ABC transporter ATP-binding cassette domain, AAA ATPase domain, Adenylylsulphate kinase domain and others. Similarly, RGAugury extracted 13 such false positives. By contrast, the 8 false positives extracted by RRGPredictor are RPW8-containing proteins lacking an NB-ARC domain. In conclusion, the NLRtracker tool we developed here is more sensitive and more accurate than previously available tools for extracting NLRs from a given plant proteome/transcriptome. Additionally, NLRtracker facilitates domain architecture analysis and phylogenetic analysis. Combining the extracted NB-ARC domain generated by NLRtracker with the RefPlantNLR extracted NB-ARC dataset (**S9 Dataset**) should greatly facilitate comparative phylogenetics and reveal

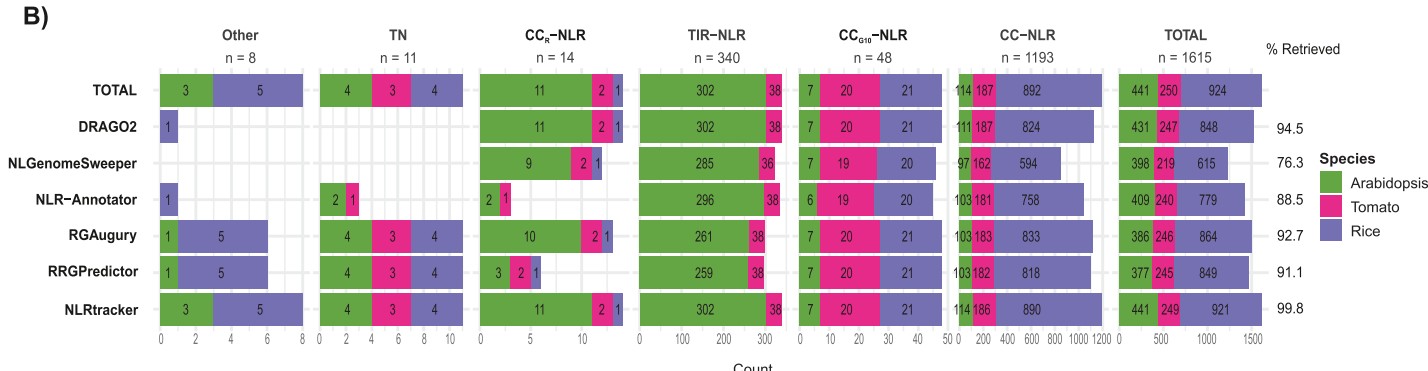

**Fig 6. NLRtracker is the most sensitive and accurate NLR extraction tool on the *Arabidopsis*, tomato, and rice RefSeq genomes.** Benchmarking of NLR annotation tools using the *Arabidopsis*, rice, and tomato RefSeq genomes. (**A**) NLRtracker pipeline. InterProScan and predefined NLR motifs are used to group sequences into different categories. (**B**) Number of NLRs retrieved in each NLR subclass per species. Underlying data and R code to reproduce the figures in **S3 Appendix**. CC, coiled-coil; LRR, leucine-rich repeat; NB-ARC, nucleotide-binding adaptor shared by APAF-1, certain R gene products, and CED-4; NLR, nucleotide-binding leucine-rich repeat; TIR, Toll/interleukin-1 receptor.

**Table 2. Extraction of NLRs from the *Arabidopsis*, tomato, an d rice RefSeq proteomes.**

| | | *Arabidopsis*/tomato/rice (*N* = 1,615) | |
| --- | --- | --- | --- |
| *Tool* | *Input* | *Sensitivity* | *Specificity**[*] |
| *DRAGO2* | AA/transcripts | 94.5% | 94.4% |
| *NLGenomeSweeper* | Transcripts/Genomic | 76.3% | 100% |
| *NLR-annotator* | Transcripts/Genomic | 88.4% | 100% |
| *RGAugury* | AA | 92.6% | 99.1% |
| *RRGPredictor* | AA/transcripts | 91.1% | 99.5% |
| *NLRtracker* | AA/transcripts | 99.8% | 100% |

[*]Percentage of retrieved sequences being genuine NLRs.

NLR, nucleotide-binding leucine-rich repeat.

the phylogenetic relationships of a newly annotated NLR. Nevertheless, the quality of the output remains dependent on the quality of the input sequences, and none of these tools can determine whether an extracted sequence represents a genuine NLR, as in having a genuine NB-ARC domain or consisting of a full-length protein. For example, 94/1,615 extracted proteins do not get annotated with an NB-ARC domain, of which 44 do not get annotated with a P-loop containing NTPase domain but do contain NB-ARC–specific motifs in combination with domains found in NLRs. Some of these may represent genuine NLRs such as Pb1 or RXL, which have undergone regressive evolution, whereas other may be partial or pseudogenes. Finally, all NLR extraction tools require independent annotation of gene models. Unfortunately, it remains difficult to correctly predict NLR gene models in an automated way, and such annotation often requires manual curation. The functional annotation of NLR gene models generated by NLRtracker can be used to assess whether a given NLR gene model is likely to be correct, or whether it lacks key features, indicating that it is either degenerated or pseudogenized, or alternatively incorrectly annotated.

## Additional applications of the RefPlantNLR dataset

We showed that RefPlantNLR is useful for benchmarking and improving NLR annotation tools. Additional uses of the dataset include providing reference points for newly discovered NLRs with NLRtracker feeding into the large-scale phylogenetic analyses that are necessary for classifying NLRomes. Phylogenetic analyses would help assign NLRs to subclades and provide a basis for generating hypotheses about the function and mode of action of novel NLRs, which phylogenetically cluster with experimentally validated NLRs. This type of phylogenetic information can be combined with other features such as genetic clustering and has, for instance, proven valuable in previous work on rice and solanaceous NLRs [23,48] and for defining the $CC_{G10}$-NLR class [13].

Furthermore, known mutants and sequence variants can be mapped onto a phylogenetic framework, such as the RefPlantNLR tree (**Fig 4**). For example, the CC-NLR ZAR1 and TIR-NLR ROQ1 are bound to ATP in their activated form [49,50], whereas the TIR-NLR RPP1 is bound to ADP in its activated form [39]. RefPlantNLR has already proven useful in interpreting a feature of the recently elucidated structure of the RPP1 resistosome [39]. The authors used RefPlantNLR to determine that although most CC-NLRs contain a TT/SR motif in which the arginine interacts with ATP, a subset of TIR-NLRs contain a charged or polar substitution creating a TTE/Q motif interacting with ADP in the activated form [39]. Interestingly, a phylogenetically distinct subgroup of CC-NLRs known as the MIC1 group [41] is an exception to this rule by having a TTE/Q motif in their ADP binding pocket and thus may also

retain ADP binding when activated. This example shows how a carefully curated reference dataset like RefPlantNLR can facilitate data interpretation and hypothesis generation.

RefPlantNLR highlights the understudied plant species of NLR biology. **S4 Table** reveals that approximately 80% (48 out of 59) of the major seed plant clades recently defined by Smith and Brown [35] do not have a single experimentally validated NLR. Certain taxa have subfamily-specific contractions and expansions and hence may contain unexplored genetic and biochemical diversity of NLR function. Looking forward, combining the output of NLRtracker with RefPlantNLR may highlight understudied subgroups of NLRs even within well-studied organisms. For example, while all currently studied $CC_R$-NLRs act as helper NLRs for TIR-NLRs, it has been reported that the $CC_R$-NLR subfamily has experienced clade-specific expansions in gymnosperms and rosids, pointing to potential biochemical specialization of this subfamily in these taxa [51]. In addition, although NLRs have been reported in non-seed plants and some of these appear to have distinct N-terminal domains [11], their experimental validation is still lacking.

The RefPlantNLR dataset has inherent limitations due to its focus on experimentally validated NLRs. First, it is biased toward a few well-studied model species and crops as illustrated in **Fig 1**. Additionally, RefPlantNLR entries are somewhat redundant with particular NLR allelic series, such as the monocot MLA and spinach alpha-WOLF, being overrepresented in the dataset (**Figs 1 and 4**). These issues, notably redundancy, will need to be considered for certain applications where it may be preferable to use the reduced redundancy dataset (**S16 Dataset**).

Finally, to facilitate the use of NLRtracker, we have run NLRtracker on the current NCBI RefSeq plant genomes (**S5 Table**, **S17 Dataset**). The NCBI RefSeq genomes have been annotated with the same genome annotation pipeline, which facilitates comparisons between species. Since NLRtracker also annotates integrated domains, we looked at the distribution of integrated domains in NLRs across the plant kingdom (**S6 Fig**). Some species such as tomato completely lack NLRs with integrated domains, whereas other flowering plant species have up to 17.5% of NLRs with integrated domains. Since NLRtracker is based on RefPlantNLR, which only contains entries from flowering plant species, the functional annotation may not be as accurate on nonflowering plant species. Indeed, *Physcomitrium patens* (a moss) and *Selaginella moellendorffii* (a lycophyte) appear to have a large proportion of NLRs with integrated domains (**S6 Fig**). When looking at the types of integrated domains, these are predominantly protein kinase domains for *P. patens* and ARM repeat-type SSFRs for *S. moellendorffii* (**S7 Fig**), which likely reflect ancient lineage-specific expansions [11]. Since integrated domains are thought to be effector targets or mimics thereof which genetically integrated into NLRs, the complete set of integrated domains provides a starting point for identifying putative effector targets (**S18 Dataset**).

## Conclusions

We hope that the RefPlantNLR resource will contribute to moving the field beyond a uniform view of NLR structure and function. It is now evident that NLRs are more structurally and functionally diverse than anticipated. Whereas a number of plant NLRs have retained the presumably ancestral 3 domain architecture of the TIR/$CC_R$/$CC_{G10}$/CC fused to the NB-ARC and LRR domains, many NLRs have diversified into specialized proteins with degenerated features and extraneous noncanonical integrated domains [15,21,24]. Therefore, it is time to question holistic concepts such as effector-triggered immunity (ETI) and appreciate the wide structural and functional diversity of NLR-mediated immunity. More specifically, a robust phylogenetic framework of plant NLRs should be fully integrated into the mechanistic study of these exceptionally diverse proteins.

## Material and methods

### Sequence retrieval

RefPlantNLR was assembled by manually crawling the literature for experimentally validated NLRs according to the criteria described in the results section. NLRs are defined as having an NB-ARC and at least 1 additional domain. Where possible, the amino acid and nucleotide sequences were taken from GenBank. For some NLRs, only the mRNA has been deposited and no genomic locus information was present. When GenBank records were not available, the sequences were extracted from the matching whole-genome sequences projects or from articles and patents describing the identification of these NLRs.

### Domain annotation

Protein sequences were annotated with CATH-Gene3D (v4.3.0) [52], SUPERFAMILY (v1.75) [53], PRINTS (v42.0) [54], PROSITE profiles (v2019_11) [55], SMART (v7.1) [56], CDD (v3.18) [57], and Pfam (v33.1) [58] identifiers using InterProScan (v5.51–85.0) [31] and predefined NLR motifs [32] using the meme-suite (v5.1.1) [59]. A custom R script (**S4 Appendix**) was used to convert the InterProScan output to the final GFF3 annotation and extract the NB-ARC domain. We routinely use Geneious Prime (v20201.2.2) (https://www.geneious.com) to visualize these annotations on the sequence. The NLR-associated signature motifs/domain IDs are the following:

- Late blight resistance protein R1: PF12061

- Rx-type CC: PF18052, cd14798, G3DSA:1.20.5.4130

- RPW8-type CC: PF05659, PS51153

- TIR: PF01582, PF13676, G3DSA:3.40.50.10140, SSF52200, PS50104, SM00255

- NB-ARC: PF00931, G3DSA:1.10.8.430

- NB-ARC used for phylogenetic analysis: overlap of G3DSA:3.40.50.300, SSF52540, G3DSA:1.10.8.430, SSF46785, G3DSA:1.10.10.10, and PF00931 signatures and motif 2, 7, and 8 from Jupe and colleagues [32]

- NB-ARC–associated motifs: motif 2, 7, and 8 from Jupe and colleagues [32], corresponding to the $CC_R$/$CC_{G10}$/CC-type RNBS-D, MHD, and linker motifs of the NB-ARC domain, respectively

- LRRs: G3DSA:3.80.10.10, PF08263, PF07723, PF07725, PF12799, PF13306, PF00560, PF13516, PF13855, SSF52047, SSF52058, SM00367, SM00368, SM00369, PF18837, PF01463, SM00082, SM00013, PF01462, PF18831, and PF18805

- Other: any other Pfam, SUPERFAMILY, and/or CATH-Gene3D annotation. Additionally, we included the PROSITE Profiles signatures PS51697 (ALOG domain) and PS50808 (zinc-finger BED domain), and the SMART signature SM00614 (zinc-finger BED domain).

### Sequence deduplication

The NLR amino acid sequences were clustered using CD-HIT at 90% sequence identity (v4.8.1 [60]; Usage: cd-hit -i RefPlantNLR.fasta -o RefPlantNLR _90 -c 0.90 -n 5 -M 16000 -d 0). A custom R script (**S4 Appendix**) was used to assign representative sequences per cluster per genus, i.e., if a single cluster contained sequences from multiple genera, we assigned a representative sequence per genus. The reduced redundancy sequences are provided in **S16 Dataset**.

## Phylogenetics

The NB-ARC domain of all NLRs were extracted and deduplicated. For sequences containing multiple NB-ARC domains, the extracted NB-ARC domain was numbered according to occurrence in the protein. Sequences were aligned using Clustal Omega [61], and all positions with less than 95% site coverage were removed using QKphylogeny [62] (**S14 Dataset**). RAxML (v8.2.12) [63] was used (usage: raxmlHPC-PTHREADS-AVX -T 6 -s RefPlantNLR. phy -n RefPlantNLR -m PROTGAMMAAUTO -f a -# 1000 -x 8153044963028367 -p 644124967711489) to infer the evolutionary history using the Maximum Likelihood method based on the JTT model [44]. Bootstrap values from 1,000 rapid bootstrap replicates as implemented in RAxML are shown [64] (**S15 Dataset**). The RefPlantNLR phylogeny was rooted on the non-plant outgroup and edited using the iTOL suite (6.3) [65].

## Figures describing RefPlantNLR

The figures describing the RefPlantNLR dataset were generated using a custom R script (**S5 Appendix**).

## Benchmarking RefPlantNLR

For benchmarking using the RefPlantNLR dataset, we used DRAGO2 (DRAGO2-API) [27], NLGenomeSweeper (v1.2.0 [30]; dependencies: Python 3.8, NCBI-BLAST+ (v2.11.0+), MUS-CLE aligner (v3.8.1551), SAMtools (v1.9-50-g18be38a), bedtools (v2.27.1-9-g5f83cacb), HMMER (v3.3.1), InterProScan (v5.47–82.0), TransDecoder (v5.5.0)), NLR-Annotator [29] (dependencies: meme-suite (v5.1.1), NLR-Parser (v3) [25]), Oracle Java SE Development Kit 11.0.9), RGAugury [26] (dependencies: CViT, HMMER, InterProScan, ncoils, NCBI-BLAST+, Pfamscan, Phobius), and RRGPredictor [28] (dependencies: InterProScan) using either amino acid, CDS, and/or the extracted NLR genomic loci as an input. Since NLGenomeSweeper and NLR-Annotator only accept nucleotide input, while RGAugury only accepts amino acid input, we only used RefPlantNLR entries for which CDS was available in the direct comparison. For the domain analysis, only the TIR, RxN-type CC, RPW8-type CC, NB-ARC, and LRR domains were considered. Additionally, sequentially duplicated domains were compressed in a single annotation. A custom R script was used to generate the analysis (**S1 Appendix**).

## Description of NLRtracker

NLRtracker (**S2 Appendix**) runs InterProScan (v5.51–85.0) [31] and FIMO from the meme-suite (v5.1.1) [59] using predefined NLR-motifs [32]. An R script that depends on the Tidy-verse [66] extracts sequences containing NLR-associated domains and classifies them into different subgroups:

- NLR: containing an NB-ARC domain

- Degenerate NLR: containing RxN-type CC, late blight resistance protein R1, RPW8-type CC, or TIR in combination with a P-loop containing nucleotide hydrolase domain not overlapping with other annotations or containing a RxN-type CC, late blight resistance protein R1, RPW8-type CC, TIR, or LRR with a RNBS-D, linker, and/or MHD motif

- TX: TIR domain containing protein lacking a P-loop containing nucleotide hydrolase domain and RNBS-D, linker, and/or MHD motif

- CCX: RxN-type CC or late blight resistance protein R1 domain containing protein lacking a P-loop containing nucleotide hydrolase domain and RNBS-D, linker, and/or MHD motif

- RPW8: RPW8-type CC domain containing protein lacking a P-loop containing nucleotide hydrolase domain and RNBS-D, linker, and/or MHD motif

- MLKL: containing HeLo domain of plant-specific mixed-lineage kinase domain like proteins (PF06760; DUF1221) [67]

We did not apply additional cutoffs to the InterProScan output. For the MEME output, we filtered for hits with a score ≥60.0 and a q-value ≤0.01. Additionally, for NLR extraction using the linker and MHD motif, we applied a more stringent cutoff requiring a score ≥85.0. NLRtracker outputs the domain architecture analysis, as well as the domain boundaries. Additionally, the NB-ARC is extracted facilitating phylogenetic analysis. The current version of NLRtracker can be accessed through GitHub (https://github.com/slt666666/NLRtracker).

### Description of refplantnlR R package

The NLRtracker output can be directly used with the refplantnlR R package to visualize the domain architecture, or, alternatively, the RefPlantNLR or NCBI RefSeq NLRtracker output can be loaded. The drawing of the domain architecture analysis is based on drawProteins [68]. The current version of refplantnlR can be accessed through GitHub (https://github.com/JKourelis/refplantnlR).

### Benchmarking on *Arabidopsis*, tomato, and rice genomes

The NCBI RefSeq proteomes of *Arabidopsis* (*Arabidopsis thaliana* ecotype Col-0; genome assembly GCF_000001735.4; TAIR and Araport annotation), tomato (*Solanum lycopersicum* cv. Heinz 1706; genome assembly GCF_000188115.4; RefSeq annotation v103), and rice (*Oryza sativa* group *Japonica* cv. Nipponbare; genome assembly GCF_001433935.1; RefSeq annotation v102) were downloaded from NCBI. We used NLRtracker, DRAGO2, RGAugury, and RRGPredictor on amino acid sequences, while we used the extracted CDS from the genomic sequence as an input for NLGenomeSweeper and NLR-Annotator.

NLRs were grouped in different subclades based on phylogenetic clustering with the RefPlantNLR $CC_R$-NLR, TIR-NLR, $CC_{G10}$-NLR, and CC-NLR subgroups, while those that did not clearly fall into any of these groups but contained a TIR-domain and P-loop containing NTPase domain were classified as TN subclade members. The remainder was grouped together and classified as other. Proteins that were extracted but did not belong to the NLR subfamily were manually inspected and classified as false positives. Additionally, TIR- or RPW8-only (TX and RPW8, respectively) proteins extracted as NLRs were marked as false positives. A custom R script was used to generate the analysis (**S3 Appendix**).

### Supporting information

**S1 Fig. RefPlantNLRs do not differ from the uncharacterized NLRs in *Arabidopsis*, tomato, and rice.** Bar chart of (**A**) the distribution of *Arabidopsis*, tomato, and rice NLRs in the reference genome as compared to the RefPlantNLR entries for which a counterpart can be identified in the reference genome, or (**B**) the number of NLRs with integrated domains. (**C**) The domain architecture of the NLRs in the *Arabidopsis* reference genome as compared to the RefPlantNLR entries from the *Arabidopsis* reference genome. Letter code as in **Fig 3**. Underlying data and R code to reproduce the figures in **S5 Appendix**. CC, coiled-coil; LRR, leucine-rich repeat; NB-ARC, nucleotide-binding adaptor shared by APAF-1, certain R gene products, and CED-4; NLR, nucleotide-binding leucine-rich repeat; TIR, Toll/interleukin-1 receptor; TN, TIR-NB-ARC.
(EPS)

**S2 Fig. Comparison of NLR-Annotator and NLGenomeSweeper on CDS versus genomic input.** NLR-Annotator and NLGenomeSweeper were run on CDS or genomic input. (**A**) Domain architecture analysis of NLR-Annotator and NLGenomeSweeper run on CDS or genomic input from each RefPlantNLR entry. Only entries for which a genomic locus was available were considered ($N = 407$). (**B**) Same as (**A**) for the representative dataset ($N = 281$). Correct domain architecture is consistent with RefPlantNLR annotation, incorrect is inconsistent with RefPlantNLR annotation. Underlying data and R code to reproduce the figures in **S1 Appendix**. CC, coiled-coil; CDS, coding sequence; NLR, nucleotide-binding leucine-rich repeat; TIR, Toll/interleukin-1 receptor.
(EPS)

**S3 Fig. Benchmarking NLR annotation tools using reduced redundancy RefPlantNLR entries.** Benchmarking of NLR annotation tools using the reduced redundancy RefPlantNLR dataset for which a CDS entry was available ($N = 299$). (**A**) UpSet plot showing intersection of RefPlantNLR entries retrieved by each annotation tool. (**B**) Domain architecture analysis produced by each NLR annotation tool per NLR subclass. Correct domain architecture is consistent with RefPlantNLR annotation, incorrect is inconsistent with RefPlantNLR annotation. Other is retrieved by NLR annotation tool but not reliably classified as NLR. Underlying data and R code to reproduce the figures in **S1 Appendix**. CC, coiled-coil; CDS, coding sequence; NLR, nucleotide-binding leucine-rich repeat; TIR, Toll/interleukin-1 receptor.
(EPS)

**S4 Fig. The TN class of proteins are distantly related to all other plant NLRs including TIR-NLRs.** The tree, based on the NB-ARC domain, was inferred using an approximately Maximum Likelihood method as implemented in FastTree [47] based on the JTT model [44]. NLRs with identical NB-ARC domains are collapsed, while for those with multiple NB-ARC domains, the NB-ARC are numbered according to order in the protein. The tree was rooted on the non-plant NLR outgroup The TIR-NLR, CC-NLR, $CC_R$-NLR, and $CC_{G10}$-NLR subclades are indicated. The TN class of plant NLRs clusters outside of the 4 major plant NLR subclades. Additionally, 3 *Arabidopsis* NLRs and 3 rice NLR cluster outside of the 4 major plant NLR subclades or the TN class. CC, coiled-coil; JTT, Jones–Taylor–Thornton; NB-ARC, nucleotide-binding adaptor shared by APAF-1, certain R gene products, and CED-4; NLR, nucleotide-binding leucine-rich repeat; TIR, Toll/interleukin-1 receptor; TN, TIR-NB-ARC.
(EPS)

**S5 Fig. Sensitivity and accuracy of NLRtracker compared to other annotation tools using *Arabidopsis*, tomato, and rice RefSeq genomes.** Benchmarking of NLR annotation tools using the *Arabidopsis*, rice, and tomato RefSeq genomes. UpSet plot showing intersection of NLRs retrieved by each annotation tool. False positive annotations are marked in red. Underlying data and R code to reproduce the figures in **S3 Appendix**. NLR, nucleotide-binding leucine-rich repeat.
(EPS)

**S6 Fig. NLRome from the NCBI RefSeq genomes using NLRtracker.** NLRtracker was run on the NCBI RefSeq proteomes ($N = 119$). (**A**) The number of loci encoding NLRs and the proportion thereof containing potential integrated domains per species are plotted as a stacked bar graph. (**B**) Proportion of NLR loci containing potential integrated domains. Underlying data and R code to reproduce the figures in **S5 Appendix**. NLR, nucleotide-binding leucine-rich repeat.
(EPS)

**S7 Fig. Diversity of potential integrated domains found in NLRs extracted from the NCBI RefSeq genomes.** The number of integrated domains per species is plotted as a stacked bar graph. All potential integrated domains were deduplicated per locus. Underlying data and R code to reproduce the figures in **S5 Appendix**. NLR, nucleotide-binding leucine-rich repeat. (EPS)

**S1 Table. Description of RefPlantNLR.**
(XLSX)

**S2 Table. Description of animal, bacterial, and archaeal NB-ARC domain containing proteins.**
(XLSX)

**S3 Table. Description of NLR-associated proteins.**
(XLSX)

**S4 Table. Plant orders represented in RefPlantNLR.**
(XLSX)

**S5 Table. NCBI RefSeq genomes on which NLRtracker was used to extract NLRs.**
(XLSX)

**S1 Dataset. Amino acid sequences of RefPlantNLR entries (fasta format).** This file contains 481 amino acid sequences.
(FASTA)

**S2 Dataset. CDS sequences of RefPlantNLR entries (fasta format).** This file contains 453 CDS sequences. For 28 RefPlantNLR entries no CDS sequence could be retrieved.
(FASTA)

**S3 Dataset. Annotated genomic sequences of RefPlantNLR entries (GenBank flat file format).** This file contains 377 genomic loci containing the gene models of 396 RefPlantNLR entries.
(GB)

**S4 Dataset. Amino acid sequences of animal, bacterial, and archaeal NB-ARC domain containing proteins (fasta format).** This file contains 13 amino acid sequences.
(FASTA)

**S5 Dataset. Amino acid sequences of NLR-associated entries (fasta format).** This file contains 15 amino acid sequences.
(FASTA)

**S6 Dataset. CDS sequences of NLR-associated entries (fasta format).** This file contains 15 CDS sequences. For 1 entry, no CDS sequence could be retrieved.
(FASTA)

**S7 Dataset. Annotated genomic sequences of NLR-associated entries (GenBank flat file format).** This file contains 13 genomic loci containing the gene models of 14 NLR-associated entries.
(GB)

**S8 Dataset. Amino acid sequences of the extracted RefPlantNLR NB-ARC domains (fasta format).** This file contains 488 NB-ARC domain amino acid sequences belonging to 479

RefPlantNLR entries.
(FASTA)

**S9 Dataset. Amino acid sequences of the unique RefPlantNLR extracted NB-ARC domains (fasta format).** This file contains 406 unique NB-ARC domains amino acid sequences.
(FASTA)

**S10 Dataset. RefPlantNLR predicted LRRs (txt format).** This file contains the LRRpredictor output for all RefPlantNLR entries containing LRRs.
(TXT)

**S11 Dataset. RefPlantNLR predicted C-JID (txt format).** HMMER (v3.3.1) output of RefPlantNLR C-JID annotations.
(TXT)

**S12 Dataset. Functional annotation of the RefPlantNLR amino acid sequences (GFF3 format).** This file contains the InterProScan annotation, as well as the converted MEME output using NLR motifs, converted LRRpredictor, and converted C-JID domain annotation.
(GFF3)

**S13 Dataset. Amino acid sequences of the extracted animal, bacterial, and archaeal NB-ARC domains (fasta format).** This file contains 13 NB-ARC domain amino acid sequences, which can be used as an outgroup for phylogenetic analysis.
(FASTA)

**S14 Dataset. Clustal Omega alignment of the unique RefPlantNLR extracted NB-ARC domains and animal, bacterial, and archaeal NB-ARC domains animal, bacterial, and archaeal NB-ARC domains (PHYLIP format).** This file contains the Clustal Omega alignment of 406 unique NB-ARC domains from the RefPlantNLR dataset and 13 animal, bacterial, and archaeal NB-ARC domains with all positions with less than 95% coverage removed. RXL and *At*NRG1.3 were omitted from this alignment.
(PHY)

**S15 Dataset. NB-ARC domain phylogeny of the RefPlantNLR entries using the Maximum Likelihood method (Newick format).** This file contains the phylogenetic analysis of the NB-ARC domain of the RefPlantNLR entries using the JTT method.
(TXT)

**S16 Dataset. Amino acid sequences of the non-redundant RefPlantNLR entries (fasta format).** This file contains 303 amino acid sequences representing the nonredundant RefPlantNLR entries at a 90% identity threshold per genus.
(FASTA)

**S17 Dataset. NLRtracker output from the NCBI RefSeq proteomes (tsv format).** This file contains the output of NLRtracker on the plant NCBI RefSeq proteomes.
(TSV)

**S18 Dataset. Integrated domains found in the NCBI RefSeq proteomes NLRs (fasta format).** This file contains the amino acid sequences of the integrated domains found in NLRs identified by NLRtracker in the NCBI RefSeq proteomes.
(FASTA)

**S19 Dataset. RefPlantNLR phylogeny (PDF format).** This file contains Fig 4 in PDF format.
(PDF)

**S1 Appendix. Benchmarking using RefPlantNLR.** Scripts and data.
(ZIP)

**S2 Appendix. NLRtracker.** Scripts.
(ZIP)

**S3 Appendix. Benchmarking using *Arabidopsis*, tomato, and rice proteomes.** Scripts and data.
(ZIP)

**S4 Appendix. Scripts and data to convert annotations, extract NB-ARC domain, and assign representative entries.**
(ZIP)

**S5 Appendix. R script used to generate figures describing RefPlantNLR.**
(ZIP)

## Acknowledgments

We thank Dan MacLean for suggestions regarding the refplantnlR R package; Adeline Harant, Philip Carella, and Hiral Shah for useful comments and feedback; and Aleksandra Białas for the domain architecture illustrations.

## Author Contributions

**Conceptualization:** Jiorgos Kourelis, Toshiyuki Sakai, Sophien Kamoun.

**Data curation:** Jiorgos Kourelis.

**Formal analysis:** Jiorgos Kourelis, Toshiyuki Sakai, Hiroaki Adachi.

**Funding acquisition:** Sophien Kamoun.

**Investigation:** Jiorgos Kourelis, Toshiyuki Sakai, Hiroaki Adachi.

**Project administration:** Sophien Kamoun.

**Software:** Jiorgos Kourelis, Toshiyuki Sakai.

**Supervision:** Sophien Kamoun.

**Writing – original draft:** Jiorgos Kourelis.

**Writing – review & editing:** Jiorgos Kourelis, Toshiyuki Sakai, Sophien Kamoun.

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
