## [Editor Report · Decision Letter 0]

11 Feb 2021

Dear Sophien, 

Thank you for submitting your manuscript entitled "RefPlantNLR: a comprehensive collection of experimentally validated plant NLRs" for consideration as a Methods and Resources by PLOS Biology.

Your manuscript has now been evaluated by the PLOS Biology editorial staff as well as by an academic editor with relevant expertise and I am writing to let you know that we would like to send your submission out for external peer review.

Please re-submit your manuscript within two working days, i.e. by Feb 15 2021 11:59PM.

Kind regards,

Ines

--

Ines Alvarez-Garcia, PhD,

Senior Editor

PLOS Biology

---

## [Decision Letter · Decision Letter 1]

15 Apr 2021

Dear Sophien,

Thank you very much for submitting your manuscript "RefPlantNLR: a comprehensive collection of experimentally validated plant NLRs" for consideration as a Methods and Resources at PLOS Biology. I'm handling your paper temporarily while my colleagues Dr Ines Alvarez-Garcia is out of the office. Your manuscript has been evaluated by the PLOS Biology editors, an Academic Editor with relevant expertise, and by four independent reviewers.

You'll see that the reviewers are broadly positive about your study, but each raises a number of concerns that will need to be addressed before further consideration. These include clarification of how some of the tools work, comparison with similar resources available, better specification of the limitations, and potential conversion to an online resource. 

In light of the reviews (below), we will not be able to accept the current version of the manuscript, but we would welcome re-submission of a much-revised version that takes into account the reviewers' comments. We cannot make any decision about publication until we have seen the revised manuscript and your response to the reviewers' comments. Your revised manuscript is also likely to be sent for further evaluation by the reviewers.

We expect to receive your revised manuscript within 3 months. 

**IMPORTANT - SUBMITTING YOUR REVISION**

*Re-submission Checklist*

*Published Peer Review*

*PLOS Data Policy*

*Blot and Gel Data Policy*

Best wishes,

Roli

Roland Roberts PhD

Senior Editor

PLOS Biology

on behalf of

Ines Alvarez-Garcia, PhD,

Senior Editor,

ialvarez-garcia@plos.org,

PLOS Biology

REVIEWERS' COMMENTS:

Reviewer #1:

[identifies himself as Michael Seidl]

The research article 'RefPlantNLR: a comprehensive collection of experimentally validated plant NLRs' by Kourelis and colleagues report on a collection of experimentally validated plant NLR immune receptors. The authors exploit this resource to i) describe features of functionally characterized NLR immune receptors, and ii) to benchmark tools used to predict NLRs from plant genomes. Based on these results, the authors propose a novel NLR prediction tool, which is based on established bioinformatic tools and resources. Tracing the evolution and diversity of NLRs in plants is a prerequisite to better understand the immune system in plant model systems and to develop disease resistant crops. Thus, the here presented dataset and the novel NLR prediction tools will be very interesting resources for the plant community with high potential impact on supporting and guiding future research.

The paper is well written with clear and informative figures; see few comments for further improvement below. Accessibility of the underlying data, description of the performed analyses, and the script for NLRtracker is exemplary as all of these are available as supplementary data, as are the relevant sequences in flat files. This should make the data accessible and useful for a wide range of plant biologists. The authors could consider making these data also accessible via a publicly available online database, which could also serve as a dynamic community hub to i) submit functionally characterized NLRs, ii) retrieve complete RefPlantNLRs sets or iii) species/lineage specific subsets. Furthermore, this database could even house complete predicted NLR repertoires from species with complete genomes. This is of course out of the scope of the current manuscript but would be an incredibly useful resource for the community. 

As discussed by the authors, the description of the generic NRL features is clearly biased towards the subset of species for which functionally characterized immune receptors are available (see also comment below). Thus, the most interesting aspect of the here presented work is the application of the RefPlantNLR set to benchmark NLR predictors, which is relevant as many predictors have been only benchmarked with very few or selected species (e.g., Arabidopsis). Based on the benchmark results and especially due to the inability of most tools to reliably classify the domain architecture of NLRs (Table 1; Figure 5B), the authors propose a novel tool - NLRtracker. The authors benchmark NLRtracker alongside five other tools, and NLRtracker is performing well in terms of sensitivity and specificity. However, the authors do not explicitly access how well NLRtracker is able to correctly classify the domain architecture, one of the main reasons to develop this novel tool. Providing this additional benchmark, for which the authors could likely use the predictions in Arabidopsis that overlap with their Arabidopsis-RefPlantNLR annotations, is essential to ascertain the usability of NLRtracker and its performance in contrast to the other established tools.

Detailed comments and suggestions:

p4: 'To validate the recovered sequences…' � could the authors please indicate how many sequences were in their initial set? 

p4: 'In addition to the 442 NLRs present…' � How many non-plant sequences or sequences with additional features were later added to the dataset?

Figure 1: For non-experts, it would be helpful to display a representative protein domain architecture for the four subclades discussed in Figure 1. Furthermore, it would be instructive if the authors would provide higher level taxonomic information for the plant species shown in the phylogenetic trees; for example, highlight monocots and dicots or the different plant clades.

p5: 'In total, 31 plant genera representing 11 taxonomic orders are listed…' � It doesn't seem to be surprising that functional characterization of NLRs has been largely focused on a few model and crop species and thus does not represent the plant biodiversity. How does the focus on a small subset of plant biodiversity impact the authors' (and others') approaches to predict and describe NLRs? For example, what can be learnt from the size distributions of NLRs based on this small subset? Could the authors speculate how this limitation could be overcome in the future?

p8: 'We selected the 5 most popular…' � How did the authors define 'most popular' in this context? Could the authors add a brief explanation on how the five tools differ and extend the description that is already provided?

p11: 'In addition to the four main subclades of NLRs, we…' � The authors report an additional TIR-NB-ARC (TN) class and note that this class clusters separately in a phylogenetic analysis. It is unclear if thus phylogenetic analysis in Meyers et al. 2002 or if it is part of the research reported here.

p12: How do the authors define 'genuine NLR' in the context of their benchmarking? Related, to determine specificity, one needs to obtain false positive calls but how these are defined based on the genuine NLRs is not clear. For example, the authors mention 'These false positives were predominantly proteins containing a P-loop containing nucleoside triphosphate hydrolase domain unrelated to the NB-ARC domain.' How did the authors determine that the P-loop domain was unrelated to the NB-ARC domain? 

Table 1: The authors should add the respective references to each tool to the table

NLRTracker: The developed pipeline relies on identification of known sequence motifs or profiles (i.e., PFAM domains) in the predicted proteomes. This process typically involves setting cutoffs to distinguish true positive from false positive matches, and thus influence the number of identified NLRs and quality of these predictions. The authors need to define which cutoffs they applied (for instance in InterproScan) and if identical cutoffs were applied for each domain or if domain specific cutoffs that reflect diversity within a domain have been used. This might also be related to potential false positives discussed above. Similarly, do the authors apply any length related cutoffs, e.g., in Fig 3C some NLRs have very small NB-ARC domains, to retrieve and classify sequences into NLRs.

Reviewer #2:

[identifies himself as Bingyu Zhao]

In this manuscript, the authors described a plant NLR database (RefPlantNLR) with 442 NLRs that have been experimentally validated. Five NLR-annotation tools were benchmarked by using the RefPlantNLR database. DRAGO2 is the most sensitive tool for the identification of NLRs. However, its annotation specificity is low. The other tools also have pros and cons. Therefore, the authors decide to develop a new pipeline, NLRtracker, for extraction and annotation of plant NLRs. Comparing to other tools, NLRtracker has significantly improved both sensitivity and specificity for extraction and annotation of plant NLRs. The authors also provide all curated datasets and the scripts used to analyze the dataset. 

The RefPlantNLR database and the NLRtracker will be a valuable resource for the plant immunity research community, and it is likely to be heavily cited in the future!

The whole experiment was well designed; the data was analyzed with appropriate bioinformatics tools and logically interpreted. The manuscript was very well prepared. I feel it is ready to be accepted for publication!

Two minor suggestions:

Fig3c, it looks like there were 3 kinds of NB-ARC domains. Please add the information in the figure legend. If they are referring to the description on page 15, the authors can add a few sentences to refer to figure 3c.

Page 8, following "that NLR-Annotator, delete an extra space

Reviewer #3:

[identifies himself as Detlef Weigel]

PLoS Biology PBIOLOGY-D-21-00318_R1

I apologize for the time it has taken me to review this work, but things are currently unpredictable.

The current work makes a very solid contribution to the exciting field of (plant) NLR biology. The authors have been extremely careful to compile an excellent set of NLR sequences from genes that have been shown to have some sort of function (overwhelmingly, conferring disease resistance) in different plant species. The collection is currently biased towards A. thaliana, but it is a living collection of sequences and I have full confidence that the authors will continuously update it, and that this bias will soon disappear, as positional cloning is quickly becoming routine even in difficult crop species.

My major concern is that the value of the resource is limited because it seems to consist primarily of downloadable flat files, instead of an interactive database that can be used to explore domain structures and sequence similarities. I would strongly urge the authors to build such a resource. 

There is not much to criticize regarding the presented data themselves, as the analyses are straightforward (even if they involved a very considerable amount of work). However, my opinion is that more could be done with the dataset without too much extra effort, and that such additional analyses would make the study considerably more appealing.

1. An important question in plant NLRology is how many of the NLRs have a bona fide function, and how many are a just byproduct of rampant sequence diversification. The authors can now ask whether the RefPlantNLR set is a random subset of annotated NLRs in the respective species, at least for the four top species (Arabidopsis, tomato, rice, wheat), or whether the RefPlantNLR set has properties that sets them apart.

2. Another related question is the population frequency of NLR alleles with likely identical function. For Arabidopsis, a collection of NLR genes and alleles from dozens of strains has been published, and the authors can now ask both whether the distribution of orthogroups defined by RefPlantNLR members across these strains is different (or not) from random NLRs, and whether the sequence variation within the RefPlantNLR orthogroups is significantly different from NLRs without known function. Perhaps one can use data from the recent Prighozin and Krasileva paper for this purpose. 

I have two further suggestions/criticisms. The first one is whether genes/alleles that are only defined by autoimmunity including hybrid necrosis should be included as RefPlantNLRs. So far, at least for genes with induced autoimmune alleles, we only know that they can be mutated in a way that they become spontaneously active - but wouldn't this likely apply also to many other NLRs? I admit, there might be something special about these genes, because these, and not other genes, showed up in mutant screens.

A more important criticism concerns the NLR annotation tools. All of these require independent annotation of gene models to derive the final NLR genes, regardless of whether they use CDS or genomic sequences as inputs. Deriving correct gene models for NLR genes is difficult, and often requires considerable manual curation. This should at least be clearly discussed, including perhaps how NLR annotation tools including the new one introduced here can potentially be used to address these difficulties.

Reviewer #4:

I come in at the first revision stage as a new reviewer. This paper is of interest to the community of plant pathology and likely more broadly to plant biology. I do not think it has major appeal outside these areas. Similar work e.g. NLRannotator http://www.plantphysiol.org/content/183/2/468 have been published elsewhere.

Here are my concerns and questions (it's a shame that line numbers are missing):

* NLRtracker works on what level? Identified loci? CDS and protein? This is not clear on the github page and in the abstract. The github page also lacks a reuse license. Is it an extractor or annotator? I see the figure 6 shows it works on transcripts/AA sequences. I think this should be clarified up front. I think the field would really benefit from a pipeline that extracts loci from raw genomic sequence, annotates gene models on these with a focus on NLRs, and functionally annotates the resulting protein sequences as NLRtracker does. This would be important to standardize the whole annotation pipeline as the diversity analysis between papers and species falls already flat if not all NLR gene models are pulled out in the first place. This is not a required for the authors to design this pipeline.

* The author should be more careful in what context they use annotation e.g. genome annotation with genes or functional annotation of proteins. This will make reading the manuscript easier. Later on the author use the term NLR-retrieval. Consistent usage of terms throughout the text would be great and really help the flow.

* Paragraph: 

"These various tools use pre-defined motifs to

classify sequences as NLRs, but they differ in the methods and pipelines. NLR-Annotator -an

extension of NLR-Parser-and NLGenomeSweeper, can also use unannotated genome

sequences as input to predict the genomic locations of NLRs (Steuernagel et al., 2020; Toda et

al., 2020). This output then requires manual annotation to extract the final gene-models and

some of the annotated loci may represent partial or pseudogenized genes. "

It is not correct that one has to manually annotate these loci. One can run gene prediction tools on extended identified loci such as braker etc. 

* The title paragraph headers could be more descriptive. 

* It would be nice to have a table in the text that clearly provides all domains (in whatever combination) are required to be found in a protein to call it a NLR. It is a bit confusing from the text. I see it is added to the methods section somewhat and it could be clearer.

* Figure 3 a and b: What do all the letter codes below the graph mean?

* It would be worthwhile to compare the methods of functional annotation for all the annotation tools bench marked in this manuscript. Also this work does only benchmark falls negative and not false positive rates. This might be useful to know as well.

* How is domain prediction accuracy defined? This is unclear. Overall the whole benchmarking section is a bit confusing as not all the tools do the same e.g. NLR-annotator does only loci and rough motifs but does not identify gene models on these loci. How does this compare in the annotation specificity with others which work on protein sequences? Also it is unclear why it is split in two section with one with and one without NLRtracker. The whole benchmarking section will benefit from a restructure for clarity. 

Overall the work seems well done (while wordy and difficult to follow at times) and contributes to the advancement of the field.

---

## [Decision Letter · Decision Letter 2]

26 Aug 2021

Dear Sophien,

Thank you for submitting your revised Methods and Resources entitled "RefPlantNLR: a comprehensive collection of experimentally validated plant NLRs" for publication in PLOS Biology. I have now obtained advice from two of the original reviewers and have discussed their comments with the Academic Editor. 

Based on the reviews (attached below), we will probably accept this manuscript for publication, provided you satisfactorily address the remaining point raised by Reviewer 1. Please also make sure to address the policy-related requests stated below.

In addition, we would like you to consider a suggestion to improve the title:

"RefPlantNLR is a comprehensive collection of experimentally validated plant disease resistance proteins from the NLR family"

We expect to receive your revised manuscript within two weeks. 

*Published Peer Review History*

*Early Version*

Best wishes,

Ines

--

Ines Alvarez-Garcia, PhD,

Senior Editor,

ialvarez-garcia@plos.org,

PLOS Biology

Fig. 1B-E; Fig. 2A-D; Fig. 3A, B; Fig. 5A; Fig. S1A-C; Fig S4; Fig. S6A, B and Fig. S7

Reviewers' comments:

Rev. 1:

The authors addressed my main concerns and comments raised by the initial submission. I am particularly happy that the authors packaged some of the visualisation capacities into a R package, which has been made accessible to the community via github. I can follow the authors' arguments to provide the raw data as flat files rather than an interactive database given the cost and efforts associated with the development and maintenance of such as resource. Thus, I have no further comments/concerns or concerns related to the analyses. However, I would like to encourage the authors to adjust the font size in the phylogenetic tree (Figure 4) to ensure readability of sequence and motif names. Furthermore, it might be helpful to highlight the species for each sequence in the phylogenetic tree.

Rev. 3:

I thank the authors for diligently responding to my suggestions. This will be a landmark publication for the plant immunity field.

---

## [Editor Report · Decision Letter 3]

23 Sep 2021

Dear Sophien,

On behalf of my colleagues and the Academic Editor, Xinnian Dong, I am pleased to say that we can in principle offer to publish your Methods and Resources paper entitled "RefPlantNLR is a comprehensive collection of experimentally validated plant disease resistance proteins from the NLR family" in PLOS Biology, provided you address any remaining formatting and reporting issues. These will be detailed in an email that will follow this letter and that you will usually receive within 2-3 business days, during which time no action is required from you. Please note that we will not be able to formally accept your manuscript and schedule it for publication until you have made the required changes.

PRESS

Best wishes,

Ines

--

Ines Alvarez-Garcia, PhD 

Senior Editor 

PLOS Biology
